# Comparison between Multiple Regression Analysis, Polynomial Regression Analysis, and an Artificial Neural Network for Tensile Strength Prediction of BFRP and GFRP

**DOI:** 10.3390/ma14174861

**Published:** 2021-08-26

**Authors:** Younghwan Kim, Hongseob Oh

**Affiliations:** Department of Civil Engineering, Gyeongsang National University, Jinju 52725, Korea; orrr0213@gmail.com

**Keywords:** GFRP, BFRP, tensile strength prediction, multiple regression analysis, response surface, polynomial regression, artificial neural network

## Abstract

In this study, multiple regression analysis (MRA) and polynomial regression analysis (PRA), which are traditional statistical methods, were applied to analyze factors affecting the tensile strength of basalt and glass fiber-reinforced polymers (FRPs) exposed to alkaline environments and predict the tensile strength degradation. The MRA and PRA are methods of estimating functions using statistical techniques, but there are disadvantages in the scalability of the model because they are limited by experimental results. Therefore, recently, highly scalable artificial neural networks (ANN) have been studied to analyze complex relationships. In this study, the prediction performance was evaluated in comparison to the MRA, PRA, and ANN. Tensile strength tests were conducted after exposure for 50, 100, and 200 days in alkaline environments at 20, 40, and 60 °C. The tensile strength was set as the dependent variable, with the temperature (TP), the exposure day (ED), and the diameter (D) as independent variables. The MRA and PRA results showed that the TP was the most influential factor in the tensile strength degradation of FRPs, followed by the exposure time (ED) and diameter (D). The ANN method provided the best correlation between predictions and experimental values, with the lowest error and error rate. The PRA method applied to the response surface method outperformed the MRA method, which is most commonly used. These results demonstrate that ANN can be the most efficient model for predicting the durability of FRPs.

## 1. Introduction

The reinforcement material in reinforced concrete undergoes corrosion due to chemical factors penetrating the concrete, which can cause structural collapse. To prevent reinforcement corrosion, fiber-reinforced polymers (FRPs) have been used in buildings, precast products, and construction strengthening [1,2,3]. FRPs can be combined with carbon, aramid, glass, and basalt fibers. For example, carbon and glass fiber-reinforced polymers (CFRPs and GFRPs, respectively) have been produced and commercialized in large quantities [4,5]. Despite the outstanding properties of CFRPs, GFRPs have been preferred owing to their cost advantage [4,5,6,7,8,9]. Recently, basalt fiber-reinforced polymers (BFRPs) have been developed with lower costs and higher environmental benefits [4].

Unlike steel reinforcements, FRPs are corrosion-free. However, because an FRP is composed of a resin and fibers, there are concerns over its durability and drawbacks exist, such as chemical attacks under external conditions and manufacturing errors when employed as a construction material [9,10,11]. In the case of GFRPs used in concrete under alkaline environments, the surface between the resin and fibers undergoes damage as it reacts with the alkali ions in the high-pH alkaline environment [6]. To assess the deterioration in the durability of GFRPs in an alkaline environment, Won [12] prepared a solution containing 1.4% KOH, 1% NaOH, and 0.16% Ca(OH)_2_ to simulate an alkaline environment and assessed the strength deterioration of GFRPs in this environment at temperatures of 20, 40, 60, and 80 °C. The longer the exposure period and the higher the temperature, the greater the decrease in the tensile strength was. Elgabbas [13] assessed the durability of BFRPs in alkaline solutions at 60 °C and found that an alkaline environment significantly deteriorates their mechanical properties.

Therefore, it is important to assess and predict the durability and strength deterioration of FRPs employed in concrete under an alkaline environment to ensure the safety and durability of the structure. The durability design of reinforced concrete structures with FRPs poses challenges in terms of the performance prediction of structures, universally, because the durability of the FRP depends on the surface characteristics between the resin and fibers, degree of fiber dispersion, and resin system [14]. In addition to alkaline environment exposure, the durability and physical mechanical properties can vary significantly with the increase in the number of environmental factors, affecting the durability, which should be considered for an accurate prediction and structural application.

Various techniques have been developed to predict material performance, such as regression analysis (RA) and artificial neural networks (ANNs) [15,16,17,18,19,20]. Among the various types of regression analyses, multiple regression analysis (MRA) is used to correlate two or more independent variables and one dependent variable. Although MRA is relatively simple and quick in making predictions, its accuracy decreases with the increase in the number of independent variables [16,20]. Recently, many researchers have constructed strength prediction models for concrete by applying ANNs when independent variables are in abundance; however, the model construction process is complex [19]. Chithra [16] compared MRA and ANN models in predicting the strength of concrete containing nano-silica and copper slag. Atici [17] used MRA and ANN models for concrete strength prediction based on the characteristics of fly ash and blast-furnace slag. The authors demonstrated that ANN models outperform MRA models in terms of predictive performance. 

Recently, response surface methods (RSMs) have been used to optimize the mixing of concrete and to assess the influence of each factor (e.g., the ratio of cement, aggregates, and water) [21,22,23,24]. Alyamac [21] proposed an optimization model using an RSM to develop eco-friendly magnetic compression concrete by recycling marble sludge. The author compared the model with experimental values in terms of accuracy and demonstrated its validity. Haque [22] used an RSM to simulate a model that can predict the desired target strength and optimize the concrete mixture, and the RSM model showed an error of less than 5%. RSM are used when performing a statistical analysis based on fitting polynomials to data and evaluating the influencing variables, i.e., independent variables and predictors. Bezerra [25] presented the advantages of applying polynomial regression in response surface analyses.

To the best of our knowledge, no predictive techniques have been used to date to predict the strength degradation of FRPs exposed to alkaline environments. Naderpour [26] trained an ANN model to predict the shear strength of concrete beams reinforced with an FRP. For the prediction, experimental data were inputted to the ANN model and compared with the calculation results of shear strength equations suggested by previous researchers by using the mean absolute error (MAE), root-mean-square error (RMSE), and mean square error (MSE) values. Zhou [27] conducted an experiment on the interfacial coupling between FRPs and concrete and presented a model with stable prediction results by simulating an ANN model using a back-propagation neural network (BPNN) method.

In this study, we analyzed the critical factors and correlations influencing the strengths of GFRPs and BFRPs exposed to alkaline environments and compared the results of three modeling approaches: MRA, polynomial regression analysis (PRA), and ANN. The independent variables affecting the strength were temperature (TP), exposure time (ED), and FRP diameter (D). Furthermore, the accuracy of each model was assessed in terms of the RMSE, correlation coefficient (R), MAE, and mean absolute percentage error (MAPE).

## 2. Analysis Methods

### 2.1. Multiple Regression Analysis

The MRA is a method for analyzing linear relationships between a dependent variable and more than one independent variable—it is an extended method of simple regression. In an MRA, the independent variables affect the dependent variable; therefore, independent variables can be established when validity is obtained in relation to the dependent variable. To describe the relationships between independent and dependent variables, the constant and regression coefficients of each variable are calculated. The general multiple regression equation is expressed in Equation (1):(1)$$Y=\alpha +\beta_{1}X_{a}+\beta_{2}X_{b}+\cdots +\beta_{k}X_{k}\pm e$$
where Y is the dependent variable, α is a constant, $\beta_{1}$–$\beta_{k}$ are the regression coefficients, $X_{1}$–$X_{k}$ are the independent variables, and e is error. The MRA model assesses the validity of the association between the independent and dependent variables using the determination coefficient, $R^{2}$. The general determination coefficient equation is given by Equation (2), and $R^{2}$ is the amount of change in the dependent variable related to the independent variables. However, the determination coefficient increases with the increase in the number of independent variables in the established model, and the adjusted determination coefficient ($R_{adj}^{2}$) can be used to assess the validity. $R^{2}$ and $R_{adj}^{2}$ are expressed in Equations (2) and (3), respectively:(2)$$R^{2}=1-\frac{\sum {\left({\hat{y}}_{i}-\bar{y}\right)}^{2}}{\sum {\left(y_{i}-\bar{y}\right)}^{2}}$$
(3)$$R_{adj}^{2}=1-\frac{n-1}{\left(n-p-1\right)\left(1-R^{2}\right)}$$
where $\sum {\left(y_{i}-\bar{y}\right)}^{2}$ is the amount of change in the dependent variable, ${\hat{y}}_{i}$ is the prediction value, $\bar{y}$ is the average of the experimental values, and $y_{i}$ is the experimental value. In Equation (3), n is the number of experimental values and p is the number of independent variables. 

### 2.2. Polynomial Regression Analysis 

The advantage of RSMs is that they have a low error and improve the prediction performance using a polynomial regression equation [28]. The polynomial regression equation is composed of a predicted variable, intersection, and square terms of the predicted variable. This study attempts to assess the accuracy of the strength prediction performance using a polynomial regression equation when applied to an RSM. The general polynomial regression equation is given by Equation (4), where Y is the predicted variable, α is a constant, $\beta_{i}$–$\beta_{ij}$ are the regression coefficients, and $X_{i}$ and $X_{j}$ are the input variables: (4)$$Y=\alpha +\sum \beta_{i}X_{i}+\sum \beta_{ij}X_{ij}+\sum \beta_{ij}X_{i}{}^{2}$$

### 2.3. Artificial Neural Network

Each neuron in the human brain is connected by tens of thousands of different neurons and billions of neurites, which operate the human nervous system by interacting with the neurons. ANNs were developed in an attempt to combine simple computational elements into complex, highly interconnected systems, and then simulate complex phenomena by modeling them into biological neural systems imitating the human brain [16,17,19,29,30]. As shown in Figure 1, an ANN comprises an input layer, a hidden layer, and an output layer, each of which contain a node called a neuron, analogous to that found in the human neural network. Early ANNs had a single-layer perceptron comprising input and output layers; however, because nonlinear separation of data is not possible in a single-layer perceptron, a multi-layer perceptron (MLP) was developed by adding hidden layers. MLPs can analyze nonlinear relationships between data with complex properties owing to the addition of hidden layers, and the technique is not much different from general statistical programs that analyze nonlinear regression and discriminant models [29].

MLPs can be learned through back propagation [16,17,29,30]. Back propagation is a method that reduces error by assigning training data to the input and calculating the error value, which is the difference between the neural network’s prediction value and the real value as a result of performing the feed-forward method and again using neural network weights. The weight sum of the back propagation is calculated using Equation (5): (5)$$x_{j}=\underset{i}{\sum }y_{i}w_{ji}$$
where $x_{j}$ is the weight value of the j-th neuron, $y_{i}$ is the input value of the i-th neuron, and $w_{ji}$ is the connective weight of the i-th to j-th neurons. The weight values are passed through a nonlinear activation function, as expressed in Equation (6):(6)$$y_{i}=\frac{1}{1+e^{-x_{j}}}$$

## 3. Model Comparison

The best analysis method for predictive models was evaluated by comparing the models developed using each analysis method. The model performance was compared using the R, RMSE, MAE, and MAPE values. The correlation coefficient (*R*) is a numerical value indicating the correlation between variables, and a linear relationship was evaluated by comparing the experimental and predicted values of the tensile strength. *R* ranges from +1 to –1, and the closer the *R* value is to 1, the better the predictive performance of the model. R is equal to the square root of the determination coefficient, as expressed in Equation (7). The RMSE is a method of measuring the prediction accuracy, which is averaged over the square of the error and then with the square roots. Since the MAE represents the average of the absolute values of the error of the predicted and experimental values, errors can be intuitively evaluated. The MAPE is a method of evaluating differences by calculating the predicted and experimental values as ratios. The following are the expressions of each indicator:(7)$$R=\sqrt{R^{2}}=\sqrt{1-\frac{\sum {\left({\hat{y}}_{i}-\bar{y}\right)}^{2}}{\sum {\left(y_{i}-\bar{y}\right)}^{2}}}$$
(8)$$RMSE=\sqrt{\frac{\sum {\left({\hat{y}}_{i}-\bar{y}\right)}^{2}}{n}}$$
(9)$$MAE=\frac{1}{n}\sum \left|{\hat{y}}_{i}-\bar{y}\right|$$
(10)$$MAPE=\frac{100}{n}\cdot \sum \frac{\left|{\hat{y}}_{i}-\bar{y}\right|}{\bar{y}}$$

## 4. Data Preparation

### 4.1. Test Specimens

The prepared FRPs were a GFRP and a BFRP based on 72% fibers and 28% vinyl ester resin [31]. Testing devices for the tensile strength of the FRPs were required because tensile test methods, such as steel rebars, are difficult to apply given that the polymer matrix may crumble [32]. Therefore, the test specimens were prepared with reference to the ACI 440.3R-12 [33] and CSA S807 [34] methods. Figure 2 shows the details of the test specimens.

### 4.2. Environmental Exposure Conditions

The deterioration of FRP is caused by alkali penetration during the pouring of concrete and hardening stages of concrete. To simulate the alkali environment, the tensile strength was assessed by exposing BFRP and GFRP for 50, 100, and 200 days by mixing a solution similar to the alkali environment of concrete. The alkali environment of concrete has pH of 12.5 to 13, and the solution is mixed as shown in Equation (11). As shown in Table 1, the temperature variables were set at 20, 40, and 60 °C, and the exposure day was at 50, 100, and 200 days. A water-proofing film was installed to prevent the evaporation of moisture.
(11)$$0.16\%\mathrm{Ca}\left({\mathrm{OH}}_{2}\right)+1\%\mathrm{Na}\left(\mathrm{OH}\right)+1.4\%K\left(\mathrm{OH}\right)$$

### 4.3. Tensile Test and Results

Tensile strength testing was conducted on a total of 69 test specimens for GFRP, except where tensile strength testing was not possible due to deterioration, and on 75 test specimens for BFRP. Table 2 shows the average tensile strength test results for the GFRP and BFRP. The tensile strength test results showed that the TP had a greater effect than the ED. In the case of exposure to an alkali solution at 20 °C, the tensile strength was not significant until 100 days of exposure; however, the tensile strength decreased sharply up to 50% after 200 days of exposure. This shows that the tensile strength decreased sharply by up to 50% when the FRPs were exposed for 200 days at a TP of 40 °C. In particular, BFRP was unable to measure the tensile strength after 200 days of exposure because of the impairment of the resin and fibers. At 60 °C exposure, the tensile strength could not be measured because of resin damage and swelling of the fibers after 50 days. The FRPs showed a nonlinear increase in the elongation at 40 and 60 °C when exposed for 200 days, and this is attributed to the damage between the resin and fiber and the fracture of fibers. 

## 5. Analysis and Results

### 5.1. MRA Results

In this study, the SPSS program was used for the MRA. A stepwise regression method was applied, in which the greater the number of variables added, the greater the number of existing analyzed variables re-analyzed, and nonsignificant variables were excluded from the analysis. The validity of the developed model was assessed through $R^{2}$, $R_{adj}^{2}$, *t*-test, F-test, and Durbin–Watson test. The developed models are expressed in Equations (12) and (13):(12)$$TS_{BFRP}=1561.9-19.4\mathrm{TP}-2.9\mathrm{ED}$$
(13)$$TS_{GFRP}=1439.7-12.99\mathrm{TP}-2.19\mathrm{ED}-17.0D$$

Table 3 shows the results of MRAs. The $R^{2}$ values for the BFRP and GFRP were 0.886 and 0.887 respectively, with a validity of more than 88%. Both models showed the same $R_{adj}^{2}$ value (0.83), which was lower than $R^{2}$. To verify the independence of the error, the Durbin–Watson values for the BFRP and GFRP were 0.863 and 0.594 respectively, indicating that the other variables increase when any one of the independent variables, namely TP, ED, and D, increases. Figure 3 shows $R_{adj}^{2}$ linear curves and scatter plots.

In MRAs, there are two methods for testing significance: the F-test and the *t*-test. The F-test examines whether the models are significant, while the *t*-test examines whether each independent variable represents a significant relationship with the dependent variable. The F-test of the BFRP was 210.86 (*p* = 0.000), and *p* was shown to be significant to less than 0.05 (5%) at the 95% confidence level. The F-test value of the GFRP, likewise, was 169.63 (*p* = 0.000), indicating a significance level of less than 0.05. The *t*-test of the two models generally showed a significant probability of 0.000; however, D of the BFRP model did not satisfy a significance level of less than 0.05, and D of the GFRP prediction model satisfied a significance level of less than 0.05. Therefore, with the TP, ED, and D of the BFRP predictive model as independent variables, D of the significance probability of 39.1% is >0.05, indicating that D does not affect the strength of the dependent variable in the BFRP, as shown in Equation (12). 

β shows the influence of each independent variable on the dependent variable, comparing the relative influence by standardizing the different unit systems of each independent and dependent variable. In the BFRP model, excluding D, the β values are −19.409 and −2.887 for TP and ED, respectively. A negative β indicates that the tensile strength decreases as the TP and ED increase. The β values of the GFRP model were −12.99, −2.19, and −17.0 for TP, ED, and D, respectively. Contrary to the GFRP, D was found to not affect the tensile strength of the BFRP.

Multicollinearity is intended to determine the correlation between independent variables when two or more independent variables exist, and the higher the correlation between independent variables, the less reliable the model parameter estimates. A multicollinearity problem arises when the highly correlated independent variable has a variance inflation factor (VIF) of 10 or more or a tolerance of 0.1 or less. The tolerance values, excluding D, of the BFRP were 0.963 for both TP and ED, and those of the GFRP were 0.997 for TP and ED and 0.998 for D. The VIF values of the BFRP were 1.039 for both TP and ED, and those of the GFRP were 1.003, 1.003, and 1.002 for TP, ED, and D, respectively. Both the BFRP and GFRP prediction models were evaluated as having no multicollinearity problems.

### 5.2. Polynomial Regression Analysis Results

The PRA was performed using the Minitab v20. To adopt the polynomial regression used in the RSM, three variables are combined to form linear, cross-product, and quadratic terms. The stepwise regression method was adopted, and ANOVA was used to determine the sequence and interactions of the significant factors. The models are expressed in Equations (14) and (15):(14)$$TS_{BFRP}=297+14.51\mathrm{TP}+10.93\mathrm{ED}+39.4D-0.2652{\mathrm{TP}}^{2}\;-0.01701{\mathrm{ED}}^{2}-0.271T\cdot \mathrm{ED}-0.311\mathrm{ED}\cdot D$$
(15)$$TS_{GFRP}=1946-29.89\mathrm{TP}+1.135\mathrm{ED}-60.2D-0.01075{\mathrm{ED}}^{2}\;-0.0352\mathrm{TP}\cdot \mathrm{ED}+1.277\mathrm{TP}\cdot D$$

The nonsignificant variables were excluded. Table 4 presents the ANOVA results. In Table 4, the highest *p*-value is the TP linear term, followed by ED. Relatively linear terms have been shown to have a significant effect on the tensile strength compared with the cross-product and quadratic terms. The $R^{2}$ values were found to have high explanatory powers of 94.78 and 93.99 in the BFRP and GFRP models, indicating that there is a good correlation between the prediction and experimental values, as shown in Figure 4.

### 5.3. ANN Analysis Results

MATLAB, which is widely used for ANN analysis in technical computing, was used in this study. The learning, validation, and testing phases in the ANN algorithm were 60%, 20%, and 20% respectively, and the number of epochs for learning the dataset was 11 for BFRP and 12 for GFRP. The Levenberg–Marquardt algorithm (LM), known to be the fastest back-propagation algorithm, was employed. In particular, the LM shows outstanding performance in nonlinear regression problems and is well-suited for mean-squared error training neural networks [34,35]. The activation function consists of a nonlinear neural network, the sigmoid function is applied to the hidden layer, and the purlin function is applied to the output layer as a linear activation function.

The TP, ED, and D values were applied to three input nodes at the input layer, one hidden layer was applied, and the tensile strength test values were applied to the output layer. It is challenging to construct an optimal model by specifying the number of nodes in the hidden layer at once. Therefore, constructing the model by appropriately selecting hidden-layer nodes is the core of the ANN model [17]. In this study, the model performance was evaluated by increasing or decreasing the number of hidden nodes and by calculating the MSE. 

As shown in Figure 5, the lowest validation MSE value of the BFRP model is 176.6 when applied with 9, and of the GFRP model is 306.7 when applied with 6. This means that the model with the lowest MSE exhibits the best performance [30]. The R values shown in Figure 6 and Table 5 for the BFRP and GFRP models are 0.994 and 0.993, respectively. Since the R value is close to 1, the model accuracy is evaluated as good.

## 6. Summary and Discussion

The importance of the factors influencing the deterioration in the tensile strength can be confirmed through the results of previous studies. Jongpil et al. [12] reported that the TP and ED had a significant effect on the tensile strength of GFRPs. In the study conducted by Elgabbas [13] on BFRPs, the TP and ED were found to be directly related to intensity degradation. In this study, the MRA showed the extent to which the TP and ED affected the tensile strength degradation, with the TP having a greater effect on the tensile strength than the ED. Furthermore, although the D was not a significant factor in the tensile strength degradation of the BFRP, it was somewhat significant in the case of the GFRP.

The MRA, PRA, and ANN methods have been used to predict the tensile strength degradation. The performance was compared using *R*, RMSE, MAE, and MAPE. Table 6 summarizes these results. By evaluating the computed values, the ANN models showed the best predictive performance. The *R* values of the BFRP were 0.94, 0.98, and 0.99 for the MRA, PRA, and ANN models respectively, and the *R* values of the GFRP were 0.94, 0.97, and 0.99, respectively. ANN models with *R* values close to 1 have been shown to perform well in prediction. Similarly, the RMSE results show that the smallest difference between the experimental and predictive values was found in the ANN models, demonstrating their superiority. When the error of each model was confirmed by MAPE, the errors in the ANN model were 3.71% and 4.68% for the BFRP and GFRP models respectively, with the error between the prediction and experimental values being less than 5%. Compared with the MRA model, there was a difference of more than three times, and compared with the PRA, there was a difference of more than two times.

These results imply that ANNs provide more reasonable performance than the MRA and PRA. Previous studies also reported similar results, confirming that the predictive performance of ANNs is satisfactory [15,16,17,18,36,37]. PRA methods also show better predictive performance than the most commonly used MRA methods. Therefore, the prediction technique used in this study utilizing an ANN method is not significantly different from those used in other studies.

## 7. Conclusions

In this study, statistical techniques were used to evaluate different factors affecting the tensile strength degradation of FRPs, and the prediction accuracy of the three most commonly used techniques to predict the tensile strength, i.e., MRA, PRA, and ANN, was compared. The TP was the most influential factor in the tensile strength degradation of the FRPs, followed by the ED. The D was not a significant factor for the BFRP, but it was significant for the GFRP. As shown in previous studies by Oh [38], it was found that erosion can easily occur at the interface between fibers and resins for GFRP, with relatively longer fiber lengths and larger diameters than BFRP.

ANN showed the best predictive performance in predicting tensile strength degradation of FPRs, followed by PRA and MRA. When creating an ANN model, it is essential to identify the appropriate number of nodes to ensure a good performance model. Predictive performance does not significantly improve if the number of nodes in the hidden layer is too large or too small. Consequently, researchers should create prediction models considering activation functions, the number of nodes, and the learning rate.

The prediction techniques examined in this study are accurate enough to predict the durability of concrete structures with FRPs. However, since the tests were conducted in a controlled environment where concrete structures were exposed to extreme conditions only, the durability of FRPs should be assessed using models that consider more diverse environmental conditions to achieve greater accuracy.

## Figures and Tables

**Figure 1 materials-14-04861-f001:**
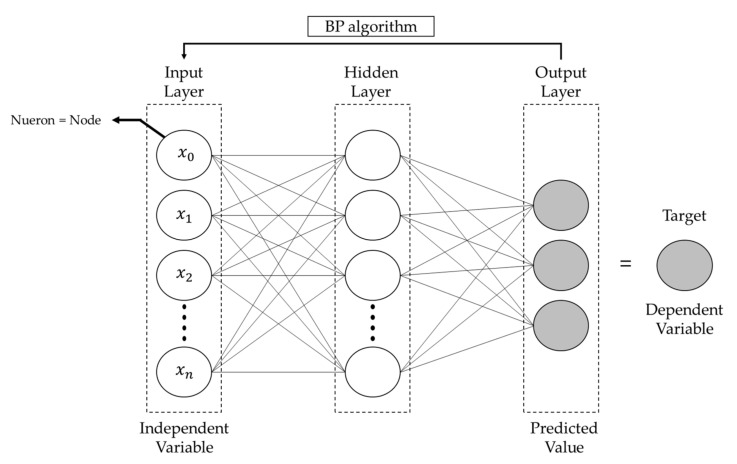
Structure of an artificial neural network.

**Figure 2 materials-14-04861-f002:**
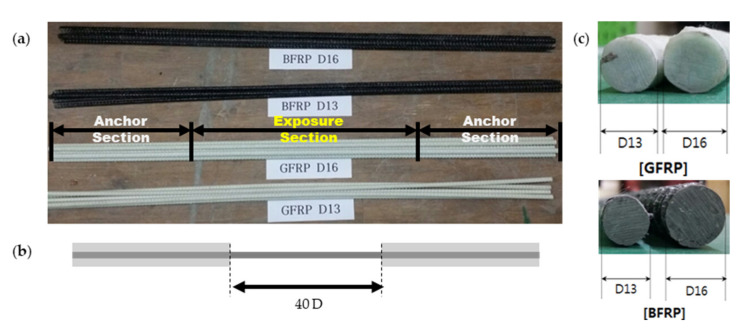
The specimens used in the test: (**a**) the shape of the GFRP and BFRP, (**b**) the length of the exposure section, and (**c**) the diameter of the GFRP and BFRP.

**Figure 3 materials-14-04861-f003:**
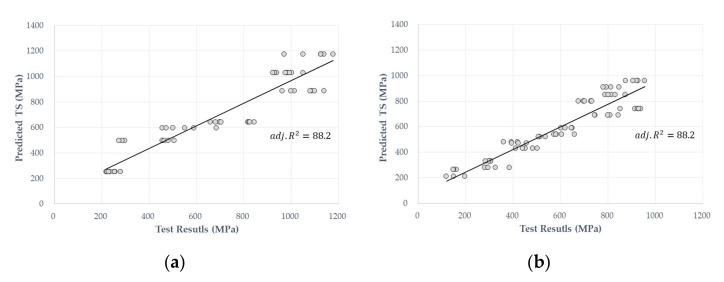
Relationship between the predictive and test values from the MRA results for the (**a**) BFRP and (**b**) GFRP predictive models.

**Figure 4 materials-14-04861-f004:**
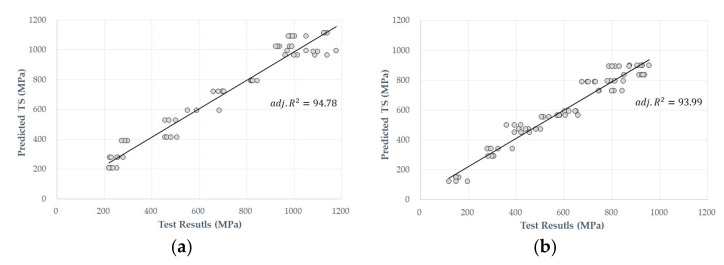
Relationship between the predictive and test values from the PRA results for the (**a**) BFRP and (**b**) GFRP predictive models.

**Figure 5 materials-14-04861-f005:**
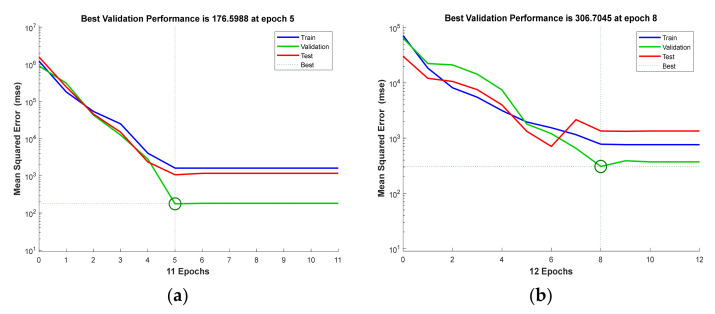
MSE training results of the training, validation, and testing stages for the (**a**) BFRP and (**b**) GFRP predictive models.

**Figure 6 materials-14-04861-f006:**
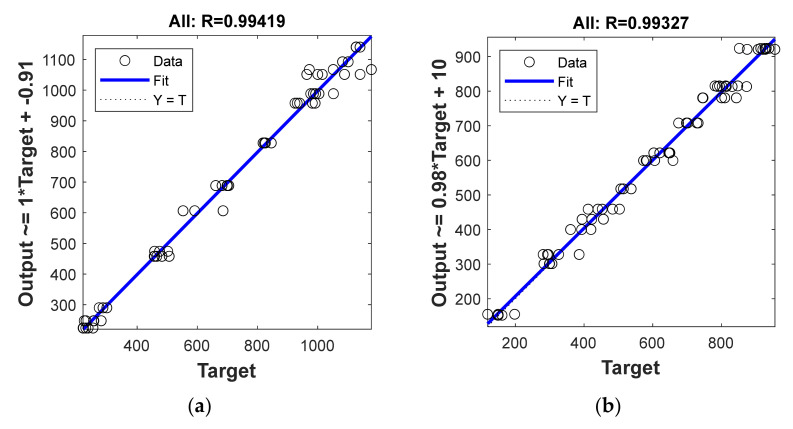
Regression plots for the relationship between the prediction and test values from the ANN results for the (**a**) BFRP and (**b**) GFRP predictive models.

**Table 1 materials-14-04861-t001:** Environmental exposure condition.

Exposure Condition	TP (°C)	ED (Days)
Reference	In air	-
Alkaline solution (pH = 12.6)	20, 40, 60	50
		100
		200

**Table 2 materials-14-04861-t002:** Tensile test results of BFRP and GFRP.

D (mm)	TP (°C)	ED (Days)	Tensile Strength (MPa) Average ± SD	
			BFRP	GFRP
13	**20**	0	1067.2 ± 84.86	917.30 ± 26.48
	20	50	953.9 ± 27.40	821.97 ± 29.39
	40	50	689.3 ± 16.02	633.80 ± 18.87
	60	50	232.4 ± 12.18	299.15 ± 8.59
	20	100	1041.7 ± 64.17	912.50 ± 31.14
	40	100	286.6 ± 10.46	391.27 ± 24.34
	20	200	609.7 ± 55.78	519.58 ± 12.90
	40	200	-	153.33 ± 5.56
16	20	0	1130.73 ± 5.61	808.95 ± 25.07
	20	50	1002.37 ± 26.49	707.16 ± 21.09
	40	50	828.40 ± 10.40	600.18 ± 31.40
	60	50	249.33 ± 20.41	316.37 ± 37.79
	20	100	1091.72 ± 9.46	789.18 ± 38.12
	40	100	473.76 ± 18.79	458.30 ± 32.11
	20	200	478.06 ± 18.08	424.37 ± 25.48
	40	200	-	155.68 ± 32.38

**Table 3 materials-14-04861-t003:** MRA results of BFRP and GFRP.

Type	Variable	β	*t*	*p*	TOL	VIF
BFRP	Constant	1561.94	34.962	0.000	-	-
	TP	−19.409	−19.491	0.000	0.963	1.039
	ED	−2.887	−10.109	0.000	0.963	1.039
	D	-	0.864	0.391	0.998	
	F(*p*)	210.862		R	0.942	
	$R_{adj}^{2}$	0.882		$R^{2}$	0.886	
	Durbin–Watson	0.863		*p*-value	0.000	
GFRP	Constant	1439.72	14.173	0.000	-	-
	TP	−12.99	−18.441	0.000	0.997	1.003
	ED	−2.19	−13.353	0.000	0.997	1.003
	D	−17.0	−2.509	0.015	0.998	1.002
	F(*p*)	169.626		R	0.942	
	$R_{adj}^{2}$	0.882		$R^{2}$	0.887	
	Durbin–Watson	0.594		*p*-value	0.000	

**Table 4 materials-14-04861-t004:** ANOVA results of the PRA.

BFRP Terms	Contribution	F-Value	*p*-Value	GFRP Terms	Contribution	F-Value	*p*-Value
TP	67.17	4.33	0.043	TP	56.14	36.89	0.000
ED	21.48	25.65	0.000	ED	31.43	5.71	0.020
D	0.16	12.25	0.001	D	1.10	25.35	0.000
${\mathrm{TP}}^{2}$	0.08	13.50	0.001	${\mathrm{TP}}^{2}$	3.76	28.85	0.000
${\mathrm{ED}}^{2}$	1.31	28.40	0.000	${\mathrm{ED}}^{2}$	0.79	9.38	0.003
TP*ED	4.65	49.31	0.000	TP*ED	1.29	14.61	0.000
ED*D	0.58	6.19	0.016	ED*D	-	-	-
R	97.68			R	97.22		
$R^{2}$	95.43			$R^{2}$	94.52		
$R_{adj}^{2}$	94.78			$R_{adj}^{2}$	93.99		
Durbin–Watson	1.265			Durbin–Watson	0.783		

**Table 5 materials-14-04861-t005:** Performance of ANN model for BFRP and GFRP.

	Training		Validation		Testing		NN Model	
	R	MSE	R	MSE	R	MSE	R	MSE
BFRP	0.991	1617.3	0.999	176.6	0.995	1069.2	0.994	1233.5
GFRP	0.993	774.2	0.996	306.7	0.990	1345.4	0.993	795.2

**Table 6 materials-14-04861-t006:** Comparison of each model developed by MRA, PRA, and ANN methods.

	MRA		PRA		ANN	
	BFRP	GFRP	BFRP	GFRP	BFRP	GFRP
R	0.94	0.94	0.98	0.97	0.99	0.99
RMSE	109.24	81.86	69.32	56.94	35.12	28.20
MAE	82.31	66.48	56.03	45.39	24.00	22.19
MAPE	13.51	14.33	9.60	8.50	3.71	4.68

## Data Availability

The data presented in this study are available on request from the corresponding author.

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
