# Peer review of "Comparison between Multiple Regression Analysis, Polynomial Regression Analysis, and an Artificial Neural Network for Tensile Strength Prediction of BFRP and GFRP"

_materials, 2021, doi:10.3390/ma14174861_

Round 1

Reviewer 1 Report

The manuscript presents a machine learning approach to estimate the tensile strength of a composite (GFRP and BFRP) under temperature and environmental conditions.

The manuscript is well organised and clearly written. Minor typos, but two full paragraphs are repeated in the manuscript (details later). The planning of the analysis is good, needing some clarifications. Post-processing of data could be improved and results could be presented in more meaningful ways. For example, since there are only two input variables of relevance for BFRP, a heat map showing EP vs TP and strength in the heat map would be more meaningful and easier to analyse. As a matter of fact, the authors never presented in a visual/graphical way what is the influence of each of the input variables on the strength. Tables and equations really don't help to understand and compare the different methods used in the manuscript.

In the conclusions, there is a lack of emphasis on what are the main conclusions of the results and manuscript. Sentences like 'Previous studies also reported similar results' do not help highlighting what is the novel contribution of this work and manuscript. There is a feeling that after a good work planning and presenting the data analysis, there is a lack of energy in presenting and interpreting the results and providing meaningful conclusions.

More specific comments/suggestions/requirements for improvement:

1. Line 107: '...the constant and beta regression...'. Missing alpha?

2. Line 130: Neural instead of nueral

3. Line 189: 200 instead of 500

4. Line 214: 'each independent variable is significant'?

5. Line 242 to 264: two all paragraphs repeated

6. Line 275: Eqs. 14 and 15?

7. Line 286 (general question): Are the learning points only those at Table 2? Looks a very small sample of points to conclude that the ANN model has had the chance to 'learn' enough. There should be a comment on this.

8. Line 289 (question): Why 10,000 epochs? Looks like an exagerated number considering there are only three independent variables (TP, EP, D).

After these points have been addressed, the manuscript can have a second reading before beeing accepted for publication.

Author Response

We appreciate reviewers’ comments and contributions. Despite our careful effort before submission of the manuscript, there are some contents to be changed and we recognized the reviewers’ comments are very significant for the contents of this study. The followings are revisions made by the authors. Some figures have been modified and changed.

Question 1. Line: 107: “’...the constant and beta regression...'. Missing alpha?”

- The sentence you pointed out describes the general meaning of constants and reaction coefficients. Because the following paragraphs describe the constant and regression coefficients of the equation, it is considered appropriate to exclude beta.

Question 2. Line: 130: “Neural instead of nueral”

- As per your comments, Spelling was modified.

Question 3. Line: 189: “200 instead of 500”

- As per your comments, “500 days” was modified.

Question 4. Line 214: “each independent variable is significant”

- I acknowledge that the sentence is unclear, according to the comments you gave me. t-test is used to evaluate the significance of variables in models that represent 95% confidence levels. Because it is possible to determine whether an independent variable significantly affects a dependent variable, the sentence was modified as follows: “In MRAs, there are two methods for testing significance: the F-test and t-test. The F-test examines whether the models are significant, while the t-test examines whether each independent variable represents a significant relationship with the dependent variable.”

Question 5. Line 242 to 264: “two all paragraphs repeated”

- Duplicate paragraphs were included in the process of editing the manual script. I corrected it because it was a mistake that I couldn't check meticulously.

Question 6. Line 275: “Eqs. 14 and 15?”

- It isn’t easy to know what you mean.

Question 7. Line 286: “Are the learning points only those at Table 2? Looks a very small sample of points to conclude that the ANN model has had the chance to 'learn' enough. There should be a comment on this”

- Table 2 shows the tensile strength of the test specimen tested for each variable as its average value. We conducted three or more tests per variable, except that due to deterioration, tensile strength tests were not possible. In the first paragraph of the 4.3 chapter, we mentioned the number of specimens tested.

Question 8. Line 289: “Why 10,000 epochs? Looks like an exagerated number considering there are only three independent variables (TP, EP, D)”

- We set 10,000 epoch in the ANN process. However, we modified the sentence because 10,000 epoch is a fixed value, and the actual epoch is BFRP 11 epoch and GFRP 12 epoch.

Reviewer 2 Report

Dear Authors,

thank you for your paper. My comments are:

  • eq. (1) - what is "e"? Not explained in the explanatory notes/description.
  • chapter 4.1. (Test specimens) - how many specimens (series) from GFRP and BFRP were prepared? Please specify the shape and dimensions of the samples. There is only a given diameter, what was the length?
  • Tab. 2 - question for all results - from how many samples were statistics made?
  • the text shows that the samples were tested only from GFRP and BFRP, which were not protected by concrete - what effect of concrete do you expect if the samples were concreted?

Best regards.

Author Response

  • Question 1. (1) - what is "e"? Not explained in the explanatory notes/description.

- The value e means an error. Added a description of e.

  • Question 2. chapter 4.1. (Test specimens) - how many specimens (series) from GFRP and BFRP were prepared? Please specify the shape and dimensions of the samples. There is only a given diameter, what was the length?

- Information on the number of specimens is added to Chapter 4.3. and we added a figure to chapter 4.1 about the details of the test specimens.

  • Question 3. 2 - question for all results - from how many samples were statistics made?

- As mentioned, we added information on the number of test specimens.

  • Question 4. the text shows that the samples were tested only from GFRP and BFRP, which were not protected by concrete - what effect of concrete do you expect if the samples were concreted?

- We conducted experiments simulating extreme environments in this study. Because the inside of real concrete is significantly different from the extreme environment applied in our experiments, it is necessary to observe durability degradation through experiments depicting the real environment. In fact, there is little movement of alkali ions inside hardened concrete, so there is no risk of severe strength degradation. However, in the event of damage such as cracks, the movement of alkali ions along with the movement of moisture can lead to degradation of FRP.

Reviewer 3 Report

Report on the manuscript

Title:  Comparison between Multiple Regression Analysis, Polyno- 2
mial Regression Analysis, and an Artificial Neural Network for 3
Tensile Strength Prediction of BFRP and GFRP 

 Authors: Younghwan Kim, Hongseob Oh

Manuscript ID: materials-1307284

Generally speaking, the manuscript is well written, the material is judiciously divided and organized and correct from scientific point of view. Some changes are, however, necessary. For these reasons I can recommend the acceptance of this paper after some corrections.

Before that the Editor makes a decision, I suggest that the authors emphasize take into account the following corrections

  1. First, the English must be improved
  2. In the Abstract must be presented what can we find in the paper and what is the original contribution. Please improve this section.
  3. In the section Conclusions will be point out the original results of the paper and will be highlight the contributions and further development.
  4. I think the authors need to emphasize more clearly the contribution of the manuscript from a scientific point of view.

If the author takes into account these observations the work can be published.

Author Response

Question 1. First, the English must be improved

- The manuscript was reviewed again to correct the error.

Question 2. In the Abstract must be presented what can we find in the paper and what is the original contribution. Please improve this section.

- we recognise that your comments are appropriate. We quantitatively evaluated the factors affecting the tensile strength degradation of FRP using statistical methods. We also present appropriate methods for predicting future tensile strength degradation of FRP by comparing statistical-based prediction techniques and ANN techniques. To make this understandable, we revised the abstract.

Question 3. In the section Conclusions will be point out the original results of the paper and will be highlight the contributions and further development.

- The conclusions also accurately summarize what we are suggesting in the paper. This paper can be useful because there have been no previous study cases that predicted a decrease in durability of FRP in a statistical manner. The prediction techniques we have studied in the future can be developed and improved in a variety of ways (such as improvements in ANN). As noted in the conclusion, we believe that predictive techniques (especially ANNs) can be a way to predict degradation of FRP and evaluate the behavior of structures.

Question 4. I think the authors need to emphasize more clearly the contribution of the manuscript from a scientific point of view.

-  It is judged that the significant effect of temperature on the tensile strength deterioration of FRP will be the effect of fiber length of FRP, resin, as in our previous study. We considered why the tensile strength was reduced by temperature, and presented what researchers should consider to develop a better predictive model (ANN).
